# The Variability of Relationship between Black Carbon and Carbon Monoxide over the Eastern Coast of China: BC Aging during Transport

Qingfeng Guo[1], Min Hu[1,2], Song Guo[1], Zhijun Wu[1], Jianfei Peng[1], Yusheng Wu[1]

[1]State Key Joint Laboratory of Environmental Simulation and Pollution Control, College of Environmental Sciences and Engineering, Peking University, Beijing 100871, China
[2]Beijing Innovation Center for Engineering Science and Advanced Technology, Peking University

*Correspondence to*: Min Hu (minhu@pku.edu.cn)

**Abstract.** East Asia is a densely populated region with a myriad of primary emissions of pollutants such as black carbon (BC) and carbon monoxide (CO). To characterize primary emissions over the eastern coast of China, a cascade of field campaigns were conducted in 2011, including the measurement of the ship cruise, island, and coastal receptor sites. The relationship between BC and CO is presented here for the first ship cruise (C1), the second ship cruise (C2), an island site (Changdao Island, CD), and a coastal site (Wenling, WL). The average BC mass concentrations were 2.43, 2.73, 1.09, 0.94, and 0.77 $\mu g \cdot m^{-3}$ for CD, WL, C1-YS (Yellow Sea), C1-ES (East China Sea), and C2-ES, respectively. For those locations, the average CO mixing ratios were 0.55, 0.48, 0.31, 0.36, and 0.27 ppm. The high loadings of both BC and CO imply the severe anthropogenic pollution over the eastern coast of China. Additionally, the linear correlation between BC and CO was regressed for each location. The slopes, i.e. the ratios of $\Delta BC$ to $\Delta CO$ derived from their relationship, were correlated well with the ratios of diesel consumption to gasoline consumption in each province/city, which reveals the vehicular emission as the common source for BC and CO and the distinct fuel structures between North and South China. The $\Delta BC/\Delta CO$ values at coastal sites (Changdao Island and Wenling) were much higher than those over Yellow Sea and East China Sea, and the correlation coefficients also showed a decreasing trend from the coast to the sea. Therefore, the quantity of $\Delta BC/\Delta CO$ and correlation coefficient are possible indicators for the aging and removal of BC.

## 1 Introduction

The atmospheric radiative forcing is caused by a variety of particulate and gaseous air pollutants. Among these particulate matters, black carbon (BC) impacts the Earth's climate directly through the absorption of the solar radiation and indirectly through its role as cloud condensation nuclei (Bond et al., 2013). The absorption induced by BC is markedly enhanced by the atmospheric oxidation and aging, as investigated by many chamber studies (Peng et al., 2016b;Guo et al., 2016;Schnaiter et al., 2005). BC aging includes the physical condensation-coagulation and the chemical oxidation which transform BC from hydrophobic to hydrophilic particles (Huang et al., 2013). It not only plays an important role on global BC distribution and

budget (He et al., 2016;Huang et al., 2013), but also has a significant influence on optical and hygroscopic properties of BC particles (Bond et al., 2006;He et al., 2015;Zhang et al., 2008;Khalizov et al., 2009a). These effects will potentially result in increasing extreme weather and weakening atmospheric circulations (Wang et al., 2013;Li et al., 2016;Wang et al., 2016). Among these gaseous pollutants, carbon monoxide (CO) is an indirect greenhouse gas through the production of ozone, methane, and carbon dioxide (Girach et al., 2014). Both of them are products of incomplete combustion of carbon-based fuels (Wang et al., 2015). Though BC and CO are from similar sources, their emission ratios vary significantly for different sources, so the variations in measured ratios can indicate the presence of different sources (McMeeking et al., 2010;Bond et al., 2004). The source-specific emission ratio is an important constraint on global climate and regional air quality model (Spackman et al., 2008).

Sources of BC colocated with CO will result in their concentration correlations, since the variances in the concentrations are affected by the same atmospheric process (Wang et al., 2011). There have been a number of studies about the relationship between BC and CO, and they show a remarkable correlation in most studies (e.g. Zhou et al., 2009;Spackman et al., 2008). They have generally been conducted at a stationary site or a cruise, while the simultaneous measurement of the both is rare. The slopes, i.e. the ratios of ΔBC to ΔCO, from the linear regressions are used to indicate different emission sources (Girach et al., 2014;Lee et al., 2013;Pan et al., 2011) and validate BC emissions from the bottom-up inventories (Wang et al., 2011;Han et al., 2009).

For BC, its atmospheric lifecycle includes emissions, transport, aging, and removal (Bond et al., 2013). The relationship between BC and CO is the result of a balance between emission sources and sinks (Spackman et al., 2008;Wang et al., 2015). Thus, differences in emission sources and removal rates (i.e. sinks) are often used to explain differences in ΔBC/ΔCO ratios (McMeeking et al., 2010). To a certain extent, the variability due to emissions and transport can be accounted for in ΔBC/ΔCO values (De Gouw and Jimenez, 2009;de Gouw et al., 2005). The atmospheric lifetime of BC is shorter than CO owing to cloud and precipitation scavenging, which results in the decreasing ΔBC/ΔCO with increasing time and distance from source. Therefore, the variations in ΔBC/ΔCO values also reflect air mass aging and wet removal processes in addition to sources (McMeeking et al., 2010).

The eastern coastal areas are the most developed in China, and are in the transport pathway of the Asian pollution outflow, especially during the East Asia monsoon in winter. The air pollutants emitted from this region and its upwind regions not only result in the deterioration of the air quality on a regional scale, but also exert an influence on downwind countries in the Pacific Rim (Feng et al., 2007;Peltier et al., 2008). In order to characterize the outflow of primary emission over the eastern coast of China, the campaigns including two cruises and two coastal sites were conducted in 2011. Among these, there was a campaign from March to April containing both the island stationary and marine cruise observation.

## 2 Measurement and Meteorology

### 2.1 Sampling sites and measurement

To characterize the outflow of the primary emission from East China, a series of campaigns were conducted in the coastal regions in 2011 (Figure 1). The first one was at Changdao Island (CD, 120.74°E, 37.92°N), Shandong province in north China from 20 March to 24 April, along with the first cruise observation (C1) conducted in Yellow Sea (C1-YS) and East China Sea (C1-ES) from 17 March to 9 April. The second one was the other cruise observation in East China Sea (C2-ES) from 28 May to 8 June. The third one was at Wenling coastal site (WL, 121.74°E, 28.43°N), Zhejiang province in south China from 1 to 28 November.

As shown in Figure 1, Changdao Island (CD) is located off the eastern coast in North China. To its west and south are the cities of Beijing and Tianjin and the provinces of Hebei and Shandong, which have the largest emission of BC in North China. Wenling (WL) is located at the eastern coast in South China. There are a lot of BC emitting at the boundary areas among Yangtz River Delta of Zhejiang, Jiangsu, and Shanghai, which will pose an impact on Wenling when the northwesterly wind is predominant. The more detailed description about these two sites can be seen in the previous studies (Guo et al., 2015;Yuan et al., 2013;Hu et al., 2013;Peng et al., 2016a). To the east of Changdao Island and Wenling is Yellow Sea and East China Sea, which are the marginal seas surrounded by China, Korea, and Japan.

A suite of online instrument was deployed for gaseous and particulate pollutants measurements during the campaigns. For the primary emission and BC aging are the focuses, both BC and CO hourly averaged data are used in this work. BC mass concentrations were continually measured by an optical attenuation technique based Aethalometer (AE-31, Magee Scientific, USA) with an integration time of 5 min. Aethalometer has been widely used for BC measurement and shown excellent agreement with other techniques such as thermal and photo-acoustic (Zhou et al., 2009;Girach et al., 2014;Nair et al., 2007;Hitzenberger et al., 2006). The uncertainty for BC mass concentration was estimated to be 10%. CO mixing ratios were measured by CO analyzer trace level enhacnced (48i-TLE, Thermo Scientific, USA) with an integration time of 1 min. The CO analyzer was calibrated using CO standard every week, and the zero checks were performed every day. The overall uncertainty for CO measurement was estimated to be less than 10%. For the cruise observation, the data with simultaniously sharp increase in concentrations of BC and CO were screened and excluded from the dataset to avoid the contamination by the ship emission.

### 2.2 Meteorological conditions

Figures 2a - 2d show the mean synoptic wind flow patterns at 925 hPa during Changdao Island (CD), the first cruise (C1), the second cruise (C2), and Wenling (WL) campaign periods, respectively, as obtained from NCEP/NCAR reanalysis (http://www.esrl.noaa.gov/psd). These flow patterns reveal the typical impact of East Asia monsoon over the eastern coast of China, which includes the winter and summer monsoon. Generally, the winter monsoon lasts from November to the

following April with the prevailing northwestern wind, while the summer monsoon continues from May to October with the predominant southwestern wind.

As can be seen in Figures 2a, 2b and 2d, Changdao Island, Yellow Sea and East China Sea, and Wenling were influenced by the winter monsoon during CD, C1, and WL campaigns, respectively, whereas East China Sea during C2 (Figure 2c) was impacted by the summer monsoon. Though CD and C1 were in the same period, C1 ended two weeks earlier than CD, as the date indicates in Figure 1. Consequently, though there was somewhat difference in the wind speed, their flow patterns were basically consistent. In addition, the wind flow pattern during C1 was also comparable with that during WL. However, there was a little discrepancy in the wind direction, which implies that they were in the opposite phases of the winter monsoon, that is, the period during C1 was at the end of the winter monsoon, which would get weaker and transit to the summer monsoon and the period during WL was at the start of the winter monsoon. The wind flow patterns during the first and second cruise were almost opposite in the wind direction (Figure 2b and 2c), which suggests that the air mass during the first cruise mainly flowed from North China to Yellow Sea, and then to East China Sea, whereas during the second cruise, the air mass direction was from South China to East China Sea.

## 3 Results and Discussion

### 3.1 Variability of BC and CO concentration

The average BC mass concentrations were 2.43, 1.09, 0.94, 0.77, and 2.73 $\mu g \cdot m^{-3}$ for CD, C1-YS, C1-ES, C2-ES, and WL, respectively. Correspondingly, the average CO mixing ratios were 0.55, 0.31, 0.36, 0.27 and 0.48 ppm. The average concentrations between coastal sites were similar, so were the concentrations between different sea areas. It is no doubt that the pollutants' concentrations at coastal sites are higher than those in the marine atmosphere, but BC and CO in Yellow Sea and East China Sea still had considerable loadings, implying the severe anthropogenic pollution from the continent.

BC and CO at Changdao Island had concentration ranges of 0.3 - 8.5 $\mu g \cdot m^{-3}$ and 0.1 - 2.9 ppm (Figure 3a), while BC at Wenling had a wider range of 0.1 - 13.7 $\mu g \cdot m^{-3}$ and CO had a narrower range of 0.1 - 1.6 ppm (Figure 3b). This difference between coastal sites is associated with the distinct pollutants emissions between North and South China, which will be discussed further in the section 3.2. Meanwhile, except the pollution episode on 8 April during C1-YS, the concentrations for BC and CO over the sea were less than 4 $\mu g \cdot m^{-3}$ and 1 ppm, respectively. The different concentration ranges between coastal sites and sea areas is related with the distance to the continental source. The episode in Yellow Sea on 8 April also occurred at Changdao Island from 7 to 8 April (shown in the dashed rectangle in Figure 3a and 3b), indicating a regional pollution episode over these areas. The peak concentrations for BC (CO) between the island and Yellow Sea were almost the same, but the peak one for Changdao Island (18:00 April 7, local time) appeared 14 hours earlier than that for Yellow Sea (8:00 April 8, local time), which could be considered as the transport time between the island and Yellow Sea during the regional pollution. In order to verify it, the forward trajectory starting at Changdao Island and the backward one starting in Yellow Sea were respectively run (http://www.ready.noaa.gov). The green line (Fig. S1 in the Supplement) is the 24 hours forward

trajectory starting at BC peak time for Changdao Island, and the green one (Fig. S2) is the 24 hours backward trajectory starting at BC peak time for Yellow Sea. They both show that the transport time from Changdao Island to Yellow Sea is about 12 hours, agreed with the peak time lag of 14 hours.

As illustrated in Figure 3, the concentrations of BC and CO fluctuated consistently over the eastern coast of China, which indicated that they were from the same source. Apparently, the consistence during CD, C1 and WL is much better than that during C2. In particular, BC and CO at Wenling site showed the best agreement during the period from 22 to 28 November, suggesting the significant impact of the primary emission on the site. The reasons for the above variability will be discussed in the next two sections.

### 3.2 ΔBC/ΔCO variability and comparison with other studies in East China

Figure 4 shows the relationship between BC and CO for all campaigns. The data points for Yellow Sea in the first cruise (C1-YS) are all overlapped with those for Changdao Island (Figure 4a), which is similar with those between East China Sea in the second cruise (C2-ES) and Wenling (Figure 4b). It indicates that both C1-YS and CD (or C2-ES and WL) were influenced by the same air mass. However, the data points within the dashed oval in Figure 4b are apart from most of the data for Wenling campaign. These data corresponded to those on 19 November (dashed rectangle in Figure 2c) when CO mixing ratio was highest during the campaign and BC mass concentration was relatively low. In the campaign document, a heavy precipitation was recorded in the midnight of 18 November. This is agreed with the different removal mechanism that the precipitation can much more easily remove aged BC without affecting CO (Hertel et al., 1995;Girach et al., 2014). So the data impacted by the precipitation are excluded in regressing the ΔBC/ΔCO slope for Wenling.

The ΔBC/ΔCO values at coastal sites are compared with those in other studies in East China (Figure 5a) to find out the possible reasons for the distinct ratios among the continental sites. The studies which simultaneously measured BC and CO are centered in the megacities such as Beijing, Shanghai, and Guangzhou (Han et al., 2009;Zhou et al., 2009;Andreae et al., 2008), while there are still rare studies in North China Plain that emits the most amounts of BC, or the ΔBC/ΔCO value is not given in the publication although BC and CO were measured (Sun et al., 2013). Since the continental sites are close to source regions, it is speculated that the ΔBC/ΔCO values are determined by the primary emission more than by the atmospheric processing.

The strong and positive correlation between BC and CO is attributed to the common sources such as vehicular emissions (Badarinath et al., 2007). In the vehicular emissions, CO is primarily emitted from gasoline vehicles while BC emissions are dominated by diesel vehicles (Han et al., 2009). In a previous study (Zhou et al., 2009), the difference in ΔBC/ΔCO values between Beijing and Shanghai has been attributable to the higher percentage of diesel vehicles in Shanghai. As shown in Figure 5a, the ΔBC/ΔCO values in Beijing and Changdao Island in North China are less than those in Nanjing, Shanghai, Wenling, and Guangzhou in South China, hinting the disparate fuel structrues in North and South China. To prove it, the ΔBC/ΔCO values at different sites are compared with the ratios of the diesel consumption to the gasoline consumption in

each province/city (China Energy Statistical Yearbook, 2013) and they show considerable correlation ($R^2 = 0.63$, Figure 5b), which confirms that BC and CO are mainly from vehicular emissions. The comsumption ratio less than 1 indicates that the gasoline is dominated in North China, while the ratio more than 1 implies that the diesel is dominated in South China. However, the data point for Changdao Island in Shandong province is excluded from the regression line for other sites. The reason is that Changdao Island is a rural site with little local vehicle emission, and it was influenced by Beijing and its surrounding regions during winter Asian monsoon when the predominant wind was from northwest (Figure 2a). The $\Delta BC/\Delta CO$ value at Changdao Island was thus less than that in Beijing.

Although the $\Delta BC/\Delta CO$ values and consumption ratios have a good correlation ($R^2 = 0.63$), the consumption ratios can not fully explain the variability of $\Delta BC/\Delta CO$ values. In one aspect, the diesel and gasoline consumption for vehicle is only a part of the total fuel consumption. In another aspect, BC and CO are not only controlled by emission from the local province/city, but also by emission transported from other areas on a regional scale. Moreover, other sources such as biomass burning can also contribute to BC and CO, and change the $\Delta BC/\Delta CO$ value.

### 3.3 BC aging during transport

The $\Delta BC/\Delta CO$ variability can result from the spatial variation of BC and CO source/sink strength (Badarinath et al., 2007). Since most of BC emission sources are centered in East China (Figure 1), the $\Delta BC/\Delta CO$ variability depends on emission sources before BC leaves the continent, as indicated by the comparison in the section 3.2 which elucidates the disparate fuel structures between North and South China. When BC transports to the marine boundary layer, the variability in the $\Delta BC/\Delta CO$ ratio is dominantly associated with BC aging and removal, given the insignificant anthropogenic sources in the marine. Therefore, the ratios between the continental and marine atmospheres may be the ideal comparison to reflect the aging extent of BC.

Owing to the aging and removal of BC and the longer atmospheric lifetime of CO, the slopes, i.e. the ratios of $\Delta BC$ to $\Delta CO$ and correlation coefficients will decrease together from upwind to downwind areas. The $\Delta BC/\Delta CO$ values for Changdao, C1-YS (excluding the episode data) and C1-ES are 4.58, 3.49, and 1.84 $\mu g \cdot m^{-3} \cdot ppm^{-1}$ respectively, showing a descent trend from north to south over the eastern coast of China (Figure 4c). It is consistent with the predominant northwestern wind during the winter monsoon (Figures 2a and 2b). Meanwhile, the correlation coefficients reduce from 0.68 to 0.28 (Figure 4c). So the slopes and correlation coefficients determined from the linear regression are possible indicators of the aging and deposition of BC during transport. It can be evidenced by the pollution episode in Yellow Sea during the first cruise, where BC and CO has a slope of 3.30 $\mu g \cdot m^{-3} \cdot ppm^{-1}$ and a correlation coefficient of 0.68. Though the slope is smaller than that at Changdao Island (4.58 $\mu g \cdot m^{-3} \cdot ppm^{-1}$), they have the same correlation coefficient.

Under the influence of the summer monsoon (Figure 2c), East China Sea is located in the downwind of Wenling. The $\Delta BC/\Delta CO$ values for Changdao Island (4.58 $\mu g \cdot m^{-3} \cdot ppm^{-1}$) and C2-ES (4.84 $\mu g \cdot m^{-3} \cdot ppm^{-1}$) are similar, but the $\Delta BC/\Delta CO$ value for Wenling (9.15 $\mu g \cdot m^{-3} \cdot ppm^{-1}$) is two times more than those at CD and C2-ES, which means that the source region

for C2-ES is in South China other than in North China. So, the campaigns of C2-ES and WL can be considered as a transport process, though these two campaigns are not simultaneously conducted. The $\Delta BC/\Delta CO$ values and correlation coefficient for WL and C2-ES during the summer monsoon also show a descent trend, as the same as those for CD, C1-YS, and C1-ES during the winter monsoon (Figure 4c). Therefore, the decreasing slopes and correlation coefficients from source to receptor areas indicate the more aging and easier removal of BC after outflow from the source regions. It is well known that in microscopic view BC aging is generally indicated by the coating thickness, and the coating thickness is associated with the mixing state and morphological variation (Khalizov et al., 2009b;Pagels et al., 2009), which ultimately enhance BC aging. It is provided here that in macroscopic view BC aging and subsequent removal result in variation of $\Delta BC/\Delta CO$ values and correlation coefficients between BC and CO, which deepens the comprehensive understanding on BC aging.

The BC average concentration for C1-ES (0.94 $\mu g \cdot m^{-3}$) during the winter monsoon was only a little higher than that for C2-ES (0.77 $\mu g \cdot m^{-3}$) during the summer monsoon. However, the $\Delta BC/\Delta CO$ value in C1-ES was 8/20 time less than that in Changdao Island, and the ratio in C2-ES was nearly 11/20 time less than that in Wenling, indicating more aging of BC in East China Sea during the winter monsoon. Due to the more extent of aging, BC during the winter monsoon could be more hygroscopic and result in more significant radiative effect (Moffet and Prather, 2009;Bond and Bergstrom, 2006).

**4 Conclusions**

The atmospheric campaigns including two island/coastal sites and two cruises were conducted in 2011 to characterize the outflow of primary emission over the eastern coast of China. Due to a large amount of continental pollutant emissions, there were considerable loadings of BC and CO in the coast and the sea areas in East China under the influence of the Asian monsoon.The slopes, i.e. the ratios of $\Delta BC$ to $\Delta CO$ from the relationship between BC and CO are regressed to deduce the information about BC source and aging during transport. The $\Delta BC/\Delta CO$ values in North China are smaller than those in South China, which reveals the disparate fuel structures between North and South China. The $\Delta BC/\Delta CO$ values are well associated with the ratios of the diesel consumption to the gasoline consumption in each province/city, which confirms that BC and CO are primarily from the vehicular emission. The comsumption ratio implies that the gasoline is dominated in North China while the diesel is dominated in South China.

The comparison in $\Delta BC/\Delta CO$ values between the coastal site and the sea area reflect the aging and deposition of BC. During the simultaneous measurement of Changdao Island and the first cruise, the $\Delta BC/\Delta CO$ value and the correlation coefficient decreased with the distance from the source under the influence of the winter monsoon. The $\Delta BC/\Delta CO$ value and the correlation coefficient also showed a decreasing trend from Wenling to East China Sea. Therefore, the $\Delta BC/\Delta CO$ ratio and the correlation coefficient are possible indicators for BC aging and removal after outflow from the source regions, which deepens the comprehensive understanding on BC aging in macroscopic view.

**The authors declare that they have no conflict of interest.**

**Acknowledgements**

This work was supported by National Basic Research Program of China (973 Program) (2013CB228503), National Natural Science Foundation of China (91544214, 41421064, 21677002), China Ministry of Environmental Protection's Special

Funds for Scientific Research on Public Welfare (201009002), and National Key Research and Development Program of China (2016YFC0202003). We thank the CAPTAIN team from Peking University, Peking University Shenzhen Graduate School, and Zhejiang Province Environmental Monitoring Center for their help and support for this research.

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

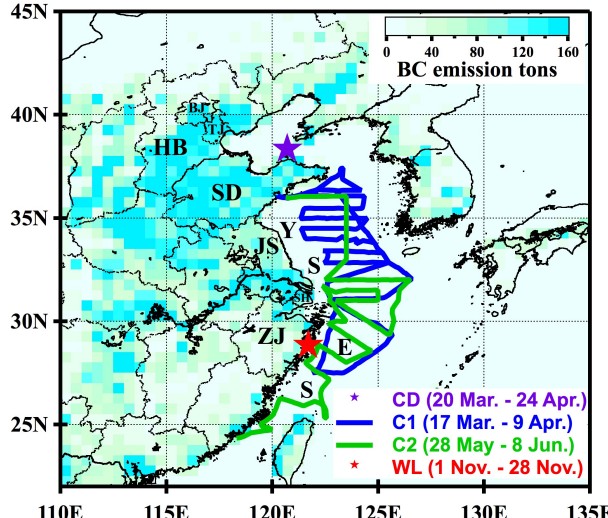

**Figure 1. The coastal sites, cruise tracks and their observation periods for the campaigns conducted in 2011. Deep purple star is the coastal site of Changdao Island (CD), and red star is the coastal one of Wenling (WL). Blue line is the track of the first cruise (C1) and green line is the track of the second cruise.The yearly mean anthropogenic emission of BC is also colored on the map (MEIC, http://www.meicmodel.org). The abbreviation for provinces/cities: BJ - Beijing, TJ - Tianjin, HB - Hebei, SD - Shandong, JS - Jiangsu, SH - Shanghai, ZJ - Zhejiang, YS - Yellow Sea, ES - East China Sea.**

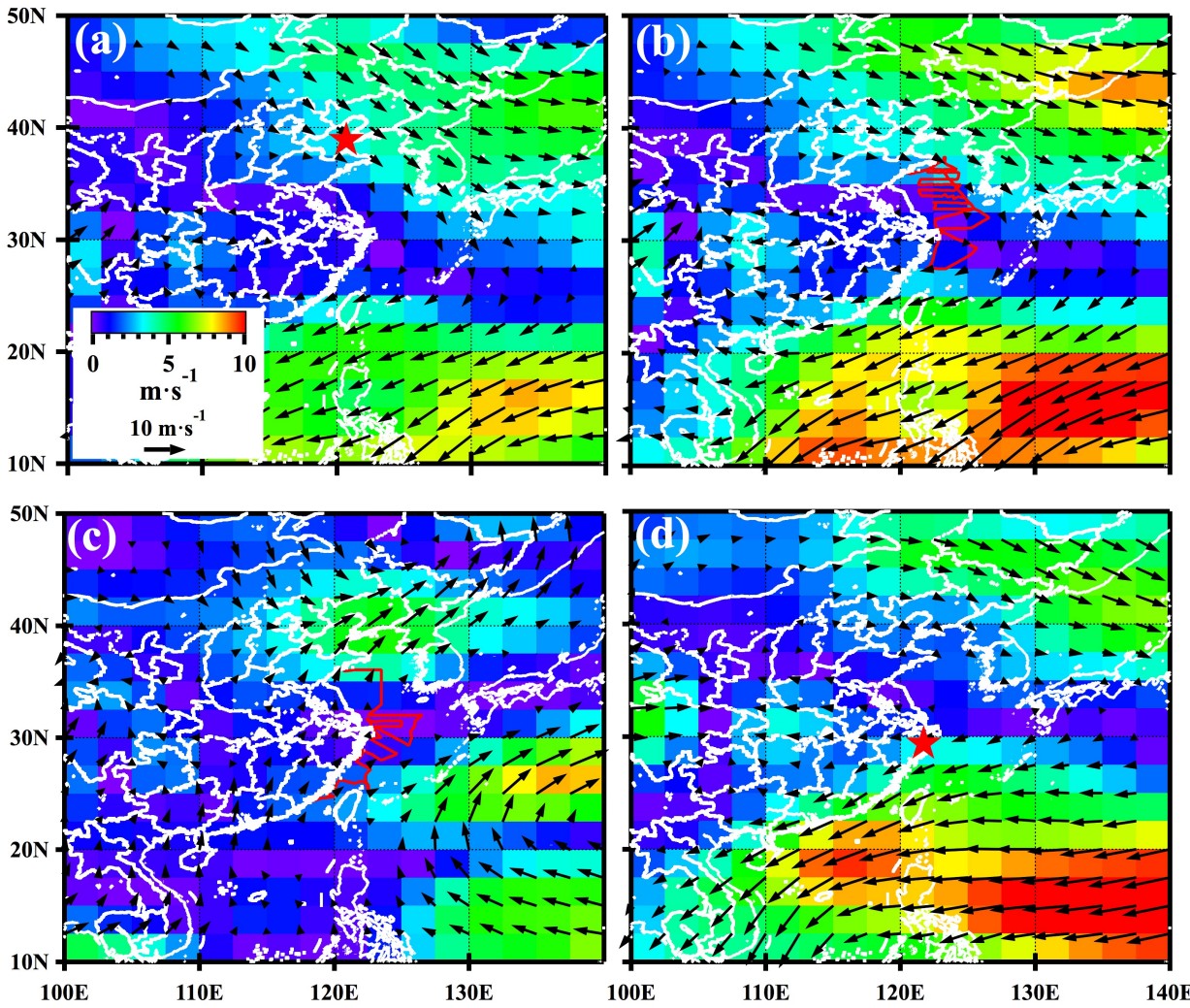

**Figure 2. The synoptic wind flow patterns at 925 hPa averaged over Changdao Island (a, the red star, 20 March - 24 April), the first cruise (b, the red line, 17 March - 9 April), the second cruise (c, the red line, 28 May - 8 June), and Wenling (d, the red star, 1 - 28 November) campaign periods as shown in Figure 1. The arrow length and the color show the wind speed, while the arrowhead indicates the wind direction.**

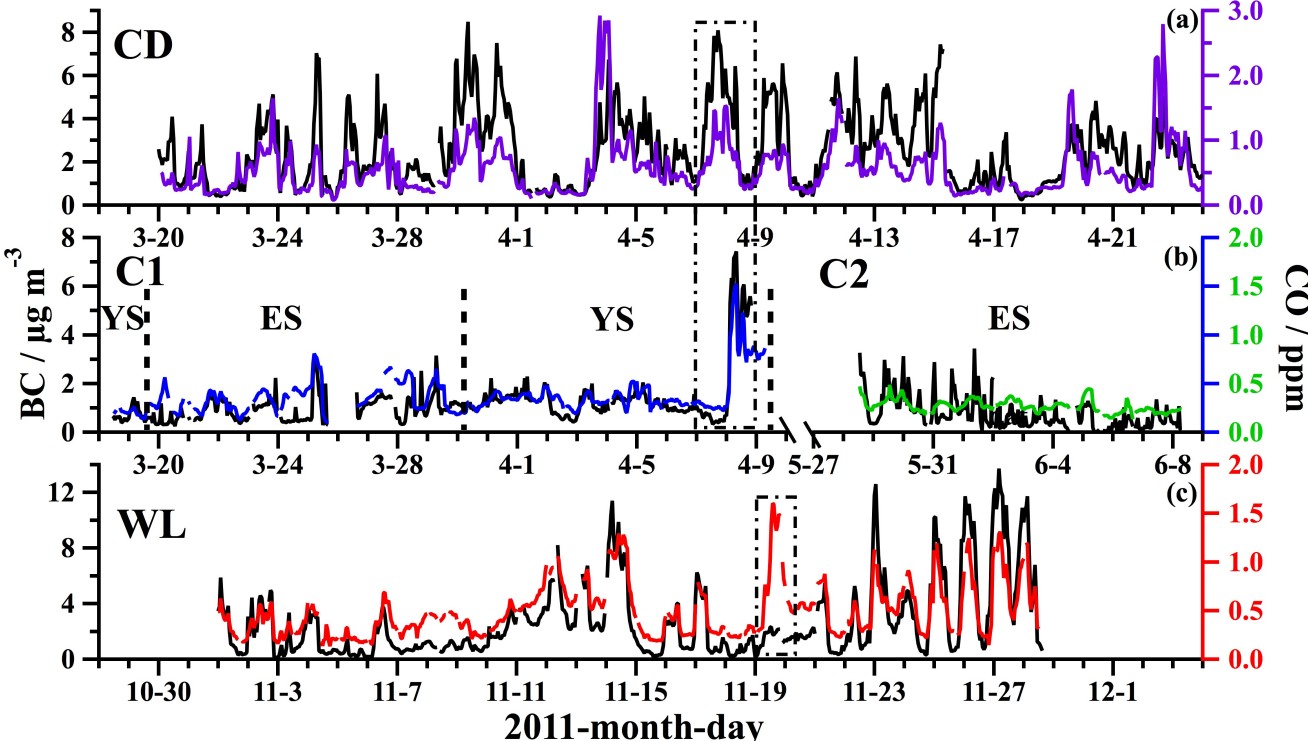

**Figure 3. The time series of BC (the black lines) and CO (the lines coded by other colors) during the campaigns of Changdao Island (a), two cruises (b), and Wenling (c).**

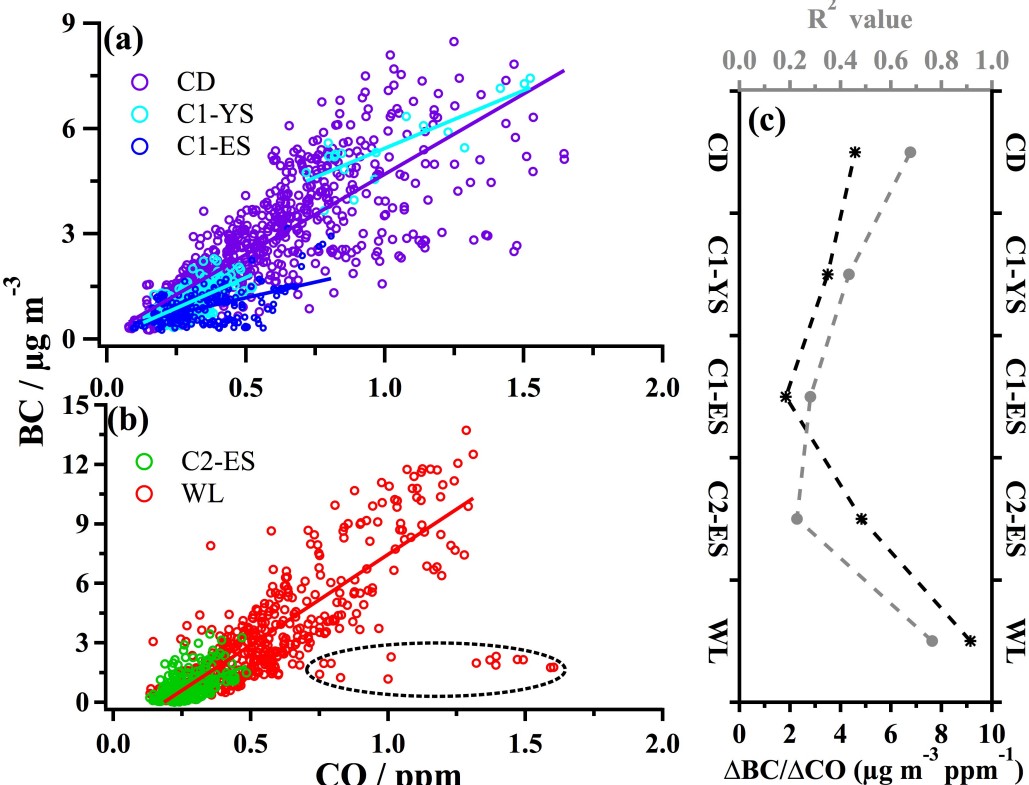

**Figure 4. Scatter plots of BC vs. CO (a, b) and their regression slopes and correlation coefficients (c).**

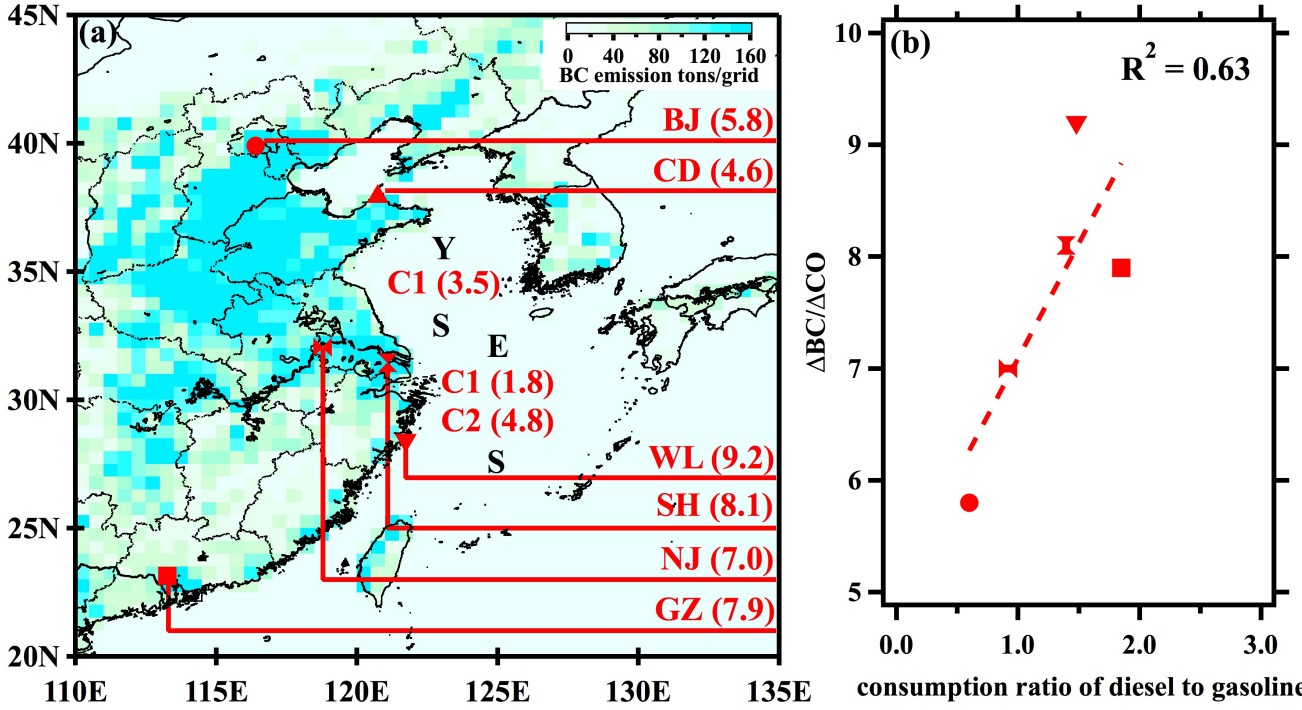

**Figure 5.** The ΔBC/ΔCO ratios in this study and other studies in East China (a) and their function of the ratios of diesel consumption to gasoline consumption in each province/city (b).