# Peer review of "The Variability of Relationship between Black Carbon and Carbon Monoxide over the Eastern Coast of China: BC Aging during Transport"

_Atmospheric Chemistry and Physics, 2017_

## Referee Comment (RC1) · Anonymous Referee #2 · 9 Apr 2017

The authors conducted comprehensive measurements of ship cruise, island, and coastal receptor sites over the eastern coast of China. They analyzed the linear relationship between BC and CO and inferred very useful information about emission sources (e.g., fuel structures) as well as BC aging and removal in continental outflows. This study can improve our understanding on BC emissions, aging, and removal over the eastern coast of China. Before the manuscript can be considered for publication, I have a few comments for the authors to address.

1. Introduction Section: Since this study is focusing on BC aging during transport, there are not enough descriptions/discussions on BC aging process, such as defining BC aging and highlighting the importance of BC aging. For example, BC aging is

commonly defined as the physical and chemical transformation of BC from hydrophobic to hydrophilic particles. BC aging significantly influences global BC distribution and budget (e.g., He et al., 2016; Huang et al., 2013) as well as BC optical properties (e.g., He et al., 2015; Bond et al., 2006), further affecting global BC radiative effects. It would be helpful if the authors could include these recent studies and add some discussions on this aspect.

References:

Bond, T. C., Habib, G., and Bergstrom, R.W.: Limitations in the enhancement of visible light absorption due to mixing state, J. Geophys. Res.-Atmos., 111, D20211, doi:10.1029/2006jd007315, 2006.

He, C., Liou, K.-N., Takano, Y., Zhang, R., Levy Zamora, M., Yang, P., Li, Q., and Leung, L. R.: Variation of the radiative properties during black carbon aging: theoretical and experimental intercomparison, Atmos. Chem. Phys., 15, 11967–11980, doi:10.5194/acp-15-11967-2015, 2015.

He, C., Li, Q., Liou, K.-N., Qi, L., Tao, S., and Schwarz, J. P.: Microphysics-based black carbon aging in a global CTM: constraints from HIPPO observations and implications for global black carbon budget, Atmos. Chem. Phys., 16, 3077–3098, doi:10.5194/acp-16-3077-2016, 2016.

Huang, Y., Wu, S., Dubey, M. K., and French, N. H. F.: Impact of aging mechanism on model simulated carbonaceous aerosols, Atmos. Chem. Phys., 13, 6329–6343, doi:10.5194/acp-13-6329- 2013, 2013.

2. Measurement Section: In terms of cruise observations, how large is the impact of emissions from the cruise used for observations? Would the samples be contaminated by emissions of the cruise itself?

3. Page 4, Lines 20-21: Is there any way to verify that the delayed one day in peak time is approximately the transport time between island and Yellow Sea? A simple and

quick way is to run the HYSPLIT model at the NOAA website to see if the air mass can be transported from island to the Yellow Sea during that specific day.

4. There are a number of English grammatical errors, e.g., Page 5, Line 4 ("much easier remove" should be "much more easily remove"); Page 5, Line 5 ("There are not outlier data" should be "There are no outlier data"); Page 5, Line 11("north China Plain that emit" should be "north China Plain that emits"). Here are just a few examples. Please double check the entire text.

5. Page 5, Line 6: It's not accurate to state that "no outliers" indicates "negligible effects of precipitation". This could simply be due to the offsetting effects of different atmospheric processes. So please re-write this sentence.

6. Page 6, Line 6: It's not accurate to say "the BC/CO ratio is only associated with BC aging and removal". I suggest using "dominantly" instead of "only".

---

## Referee Comment (RC2) · Anonymous Referee #1 · 14 Apr 2017

This study investigates the transport and aging of black carbon (BC) through assessing the relationship between BC and carbon monoxide (CO) over the eastern coast of China. It is a new angle to characterize the complicated processes associated with BC and to help constrain such a poorly understood climate forcer in the atmosphere. The in situ measurements of BC and CO from a series of field campaigns are explored. It is good to see the authors can fully exploit those valuable data in this paper. The obtained BC/CO relationships shed new light on the pollutant transport and transformation near the source regions over East Asia. I only have some minor comments for authors to address.

1) Page 4, L20. The statement "the peak time in Yellow Sea is delayed almost one day

than that at Changdao Island" is not obvious in Fig. 3. Some quantitative assessment is suggested such as lagged correlation analysis.

2) It is not clear where the diesel/gasoline consumption data come from in the study.

3) The authors attributed the outlier (Changdao Island) in Fig. 5b to the fact that it is located in the rural area. Why not exclude this data point in the plot and purely focus on the relationships in urban area? With that, we will obtain a more significant correlation.

4) In Fig. 4c, each dot of BC/CO ratio is an average over the whole sub-campaign period. However, to accurately study the BC aging during transport, it would be better to pair the fresh BC properties in the source region with the aged one in the outflow region with a proper time lag. The time lag can be decided by the transport efficiency. I am not sure if such a method can be applied in this study.

5) Fig. 2, for which days the wind fields are plotted in each panel?

6) Fig. 3, the color-coding is a little confusing. If I understand correctly, the black lines are for BC, and the other different color lines are for CO. However, the location names (CD, WL, ES, YS, etc.) are also labeled using the similar colors. Please find a better way to avoid such an ambiguity.

7) The motivation of BC study in China should be stated in a more thorough way. Recent studies about absorbing aerosol effects on extreme weather and regional climate should be discussed:

Wang, et al. "New Directions: Light Absorbing Aerosols and Their Atmospheric Impacts", Atmos. Environ., 81, 713-715 (2013)

Li, et al. "Aerosol and Monsoon Climate Interaction over Asia", 54, Rev. Geophys. (2016)

Wang, et al. "Towards Reconciling the Influence of Atmospheric Aerosols and Greenhouse Gases on Light Precipitation Changes in Eastern China", J. Geophys. Res.

Atmos. 121(10), 5878–5887 (2016)

---

## Author Comment (AC1) · 17 May 2017

We thank the referee for his/her careful and critical review of our paper. The following are our responses to the referee's comments.

1. Introduction Section: Since this study is focusing on BC aging during transport, there are not enough descriptions/discussions on BC aging process, such as defining BC aging and highlighting the importance of BC aging. For example, BC aging is commonly defined as the physical and chemical transformation of BC from hydrophobic to hydrophilic particles. BC aging significantly influences global BC distribution and budget (e.g., He et al., 2016; Huang et al., 2013) as well as BC optical properties (e.g., He et al., 2015; Bond et al., 2006), further affecting global BC radiative effects. It would

be helpful if the authors could include these recent studies and add some discussions on this aspect. References: Bond, T. C., Habib, G., and Bergstrom, R.W.: Limitations in the enhancement of visible light absorption due to mixing state, J. Geophys. Res.-Atmos., 111, D20211, doi:10.1029/2006jd007315, 2006. He, C., Liou, K.-N., Takano, Y., Zhang, R., Levy Zamora, M., Yang, P., Li, Q., and Leung, L. R.: Variation of the radiative properties during black carbon aging: theoretical and experimental intercomparison, Atmos. Chem. Phys., 15, 11967–11980, doi:10.5194/acp-15-11967-2015, 2015. He, C., Li, Q., Liou, K.-N., Qi, L., Tao, S., and Schwarz, J. P.: Microphysics-based black carbon aging in a global CTM: constraints from HIPPO observations and implications for global black carbon budget, Atmos. Chem. Phys., 16, 3077–3098, doi:10.5194/acp-16-3077-2016, 2016. Huang, Y., Wu, S., Dubey, M. K., and French, N. H. F.: Impact of aging mechanism on model simulated carbonaceous aerosols, Atmos. Chem. Phys., 13, 6329–6343, doi:10.5194/acp-13-6329- 2013, 2013.

We thank the referee for this suggestion and discuss the topics in Line 26 - 29 in Page 1 and Line 1 - 3 in Page 2 to the following: The absorption induced by BC is markedly enhanced by the atmospheric oxidation and aging, as investigated by many chamber studies (Peng et al., 2016b;Guo et al., 2016;Schnaiter et al., 2005). BC aging includes the physical condensation-coagulation and the chemical oxidation which transform BC from hydrophobic to hydrophilic particles (Huang et al., 2013). It not only plays an important role on global BC distribution and budget (He et al., 2016;Huang et al., 2013), but also has a significant influence on BC optical properties (Bond et al., 2006;He et al., 2015). These effects will potentially result in increasing extreme weather and weakening atmospheric circulations (Wang et al., 2013;Li et al., 2016;Wang et al., 2016)

2. Measurement Section: In terms of cruise observations, how large is the impact of emissions from the cruise used for observations? Would the samples be contaminated by emissions of the cruise itself?

We thank the referee for pointing this out. The data contaminated by the ship emission

were screened in our data processing. To clarify, we have revised Line 24 - 26 in Page 3 to the following: For the cruise observation, the data with simultaniously sharp increase in concentrations of BC and CO were screened and excluded from the dataset to avoid the contamination by the ship emission.

3. Page 4, Lines 20-21: Is there any way to verify that the delayed one day in peak time is approximately the transport time between island and Yellow Sea? A simple and quick way is to run the HYSPLIT model at the NOAA website to see if the air mass can be transported from island to the Yellow Sea during that specific day.

We thank the referee for this suggestion and run the HYSPLIT model at the NOAA website. we have revised Line 31 - 32 in Page 4 and Line 1 - 3 in Page 5 to the following: In order to verify it, the forward trajectory starting at Changdao Island and the backward one starting in Yellow Sea were respectively run (http://www.ready.noaa.gov). The green line (Fig. S1 in the Supplement) is the 24 hours forward trajectory starting at BC peak time for Changdao Island, and the green one (Fig. S2) is the 24 hours backward trajectory starting at BC peak time for Yellow Sea. They both show that the transport time from Changdao Island to Yellow Sea is about 12 hours, agreed with the peak time lag of 14 hours.

4. There are a number of English grammatical errors, e.g., Page 5, Line 4 ("much easier remove" should be "much more easily remove"); Page 5, Line 5 ("There are not outlier data" should be "There are no outlier data"); Page 5, Line 11("north China Plain that emit" should be "north China Plain that emits"). Here are just a few examples. Please double check the entire text.

We thank the referee for the careful and kind help with editing the English and have already examed the entire text.

5. Page 5, Line 6: It's not accurate to state that "no outliers" indicates "negligible effects of precipitation". This could simply be due to the offsetting effects of different atmospheric processes. So please re-write this sentence.

[Figure]

We thank the referee for pointing this out and delete this sentence to avoid the arbitrariness.

6. Page 6, Line 6: It's not accurate to say "the BC/CO ratio is only associated with BC aging and removal". I suggest using "dominantly" instead of "only".

we thank the referee for this suggestion and have rivesed Line 17 - 19 in Page 6 to the following: When BC transports to the marine boundary layer, the variability in the $\Delta$BC/$\Delta$CO ratio is dominantly associated with BC aging and removal, given the insignificant anthropogenic sources in the marine.

Please also note the supplement to this comment:
http://www.atmos-chem-phys-discuss.net/acp-2017-56/acp-2017-56-AC1-supplement.pdf

---

## Author Comment (AC2) · 17 May 2017

We thank the referee for his/her careful and critical review of our paper. The following are our responses to the referee's comments.

1) Page 4, L20. The statement "the peak time in Yellow Sea is delayed almost one day than that at Changdao Island" is not obvious in Fig. 3. Some quantitative assessment is suggested such as lagged correlation analysis.

We thank the referee for this suggestion and run the HYSPLIT model at the NOAA website. we have revised Line 31 - 32 in Page 4 and Line 1 - 3 in Page 5 to the following: In order to verify it, the forward trajectory starting at Changdao Island and the backward one starting in Yellow Sea were respectively run (http://www.ready.noaa.gov). The green line (Fig. S1 in the Supplement) is the 24 hours forward trajectory starting at BC peak time for Changdao Island, and the green one (Fig. S2) is the 24 hours backward trajectory starting at BC peak time for Yellow Sea. They both show that the transport time from Changdao Island to Yellow Sea is about 12 hours, agreed with the peak time lag of 14 hours.

2) It is not clear where the diesel/gasoline consumption data come from in the study.

We thank the referee for pointing this out and add the data source in Line 31 - 32 in Page 5 and Line 1 - 2 in Page 6 to the following: To prove it, the ΔBC/ΔCO at different sites are compared with the ratios of the diesel consumption to the gasoline consumption in each province/city (China Energy Statistical Yearbook, 2013) and they show considerable correlation (R2 = 0.63, Figure 5b), which confirms that BC and CO are mainly from vehicular emissions.

3) The authors attributed the outlier (Changdao Island) in Fig. 5b to the fact that it is located in the rural area. Why not exclude this data point in the plot and purely focus on the relationships in urban area? With that, we will obtain a more significant correlation.

We thank the referee for pointing this out and exclude this data point in the figure 5b.

4) In Fig. 4c, each dot of BC/CO ratio is an average over the whole sub-campaign period. However, to accurately study the BC aging during transport, it would be better to pair the fresh BC properties in the source region with the aged one in the outflow region with a proper time lag. The time lag can be decided by the transport efficiency. I am not sure if such a method can be applied in this study.

We thank the referee for this suggestion and will try this method in next study if the data in need are all available.

5) Fig. 2, for which days the wind fields are plotted in each panel?

We thank the referee for pointing this out and add the periods to the caption in Figure

2 to make it clear in Line 4 - 5 in Page 12 to the following: Figure 2. The synoptic wind flow patterns at 925 hPa averaged over Changdao Island (a, the red star, 20 March - 24 April), the first cruise (b, the red line, 17 March - 9 April), the second cruise (c, the red line, 28 May - 8 June), and Wenling (d, the red star, 1 - 28 November) campaign periods as shown in Figure 1. The arrow length and the color show the wind speed, while the arrowhead indicates the wind direction.

6) Fig. 3, the color-coding is a little confusing. If I understand correctly, the black lines are for BC, and the other different color lines are for CO. However, the location names (CD, WL, ES, YS, etc.) are also labeled using the similar colors. Please find a better way to avoid such an ambiguity.

We thank the referee for pointing this out. We label the location names with the black color in Figure 3, and make the caption more clear in Line 2 - 3 in Page 13 to the following: Figure 3. The time series of BC (the black lines) and CO (the lines coded by other colors) during the campaigns of Changdao Island (a), two cruises (b), and Wenling (c).

7) The motivation of BC study in China should be stated in a more thorough way. Recent studies about absorbing aerosol effects on extreme weather and regional climate should be discussed: Wang, et al. "New Directions: Light Absorbing Aerosols and Their Atmospheric Impacts", Atmos. Environ., 81, 713-715 (2013) Li, et al. "Aerosol and Monsoon Climate Interaction over Asia", 54, Rev. Geophys. (2016) Wang, et al. "Towards Reconciling the Influence of Atmospheric Aerosols and Greenhouse Gases on Light Precipitation Changes in Eastern China", J. Geophys. Res. Atmos. 121(10), 5878–5887 (2016)

We thank the referee for this suggestion and discuss the topics in Line 26 - 29 in Page 1 and Line 1 - 3 in Page 2 to the following: The absorption induced by BC is markedly enhanced by the atmospheric oxidation and aging, as investigated by many chamber studies (Peng et al., 2016b;Guo et al., 2016;Schnaiter et al., 2005).

BC aging includes the physical condensation-coagulation and the chemical oxidation which transform BC from hydrophobic to hydrophilic particles (Huang et al., 2013). It not only plays an important role on global BC distribution and budget (He et al., 2016;Huang et al., 2013), but also has a significant influence on BC optical properties (Bond et al., 2006;He et al., 2015). These effects will potentially result in increasing extreme weather and weakening atmospheric circulations (Wang et al., 2013;Li et al., 2016;Wang et al., 2016).

Please also note the supplement to this comment:
http://www.atmos-chem-phys-discuss.net/acp-2017-56/acp-2017-56-AC2-supplement.pdf

**Supplement:**

*Supplement of*

**The Variability of Relationship between Black Carbon and Carbon Monoxide over the Eastern Coast of China: BC Aging during Transport**

Qingfeng Guo et al.

*Correspondence to*: Min Hu (minhu@pku.edu.cn)

[Figure]

Fig. S1. The 24 hours forward trajectory starting at BC peak time of 18:00 (local time), April 7 for Changdao Island (the black star).

[Figure]

Fig. S2. The 24 hours backward trajectory starting at BC peak time of 08:00 (local time), April 8 for Yellow Sea (the red star).

---

## Author Response (AR2)

We thank the editor for his/her careful and critical review of our paper. The following are our responses to the editor's comments.

Comments to the Author:

In your manuscript, a new variable, i.e., the ratio of ΔBC to ΔCO, was introduced and proposed as a possible indicator for aging and removal of BC. However, there was little discussion either quantitatively or qualitatively on how this variable would be linked to those that are employed conventionally to characterize atmospheric BC aging. For example, BC aging is typically reflected by the coating thickness, which is related to the mixing state and morphological variation (Khalizov et al., J. Geophys. Res. 114, D05208, doi:10.1029/2008JD010595, 2009; Pagels et al., Aerosol Sci. Tech. 43, 629, 2009). Such a change under atmospheric conditions is commonly associated with variations in the optical and cloud-forming properties of BC particles (Zhang et al., Proc. Natl. Acad. Sci. USA 105, 10291, 2008; Khalizov et al., J. Phys. Chem. 113, 1066, 2009). You concluded in the abstract "Therefore, the ΔBC/ΔCO ratio and correlation coefficient are possible indicators for the aging and removal of BC", but provided little insight on how this quantity was correlated with those related to the mixing state, morphology, and optical and hygroscopic properties of BC particles, which are ultimately employed for assessing the impacts of BC particles in atmospheric models. In addition, some discussions on the plausible mechanisms leading to rapid BC aging in China would be beneficial, such as the processes detailed by Guo et al. (PNAS, 2014).

I would also recommend a careful proofread of your manuscript to improve its readability. I provide a few examples below. Abstract, line 11, change "campaign" to "campaigns". Line 17, replace "The slopes, i.e. ΔBC/ΔCO ratios derived from their relationship were" by "The slopes, i.e., the ratio of ΔBC to ΔCO derived from their relationship, were". Line 19, replace "ratios" by "values". Line 21, replace "the ΔBC/ΔCO ratio" by "the quantity of ΔBC/ΔCO".

We thank the editor for this suggestion and add the discussion in **Line 28 - 29 in Page 1**, **Line 1 - 2 in Page 2** and **Line 5 - 9 in Page 7** to the following:

**BC aging includes the physical condensation-coagulation and the chemical oxidation which transform BC from hydrophobic to hydrophilic particles (Huang et al., 2013). It not only plays an important role on global BC distribution and budget (He et al., 2016;Huang et al., 2013), but also has a significant influence on optical and hygroscopic properties of BC particles (Bond et al., 2006;He et al., 2015;Zhang et al., 2008;Khalizov et al., 2009a).**

**It is well known that in microscopic view BC aging is generally indicated by the coating thickness, and the coating thickness is associated with the mixing state and morphological variation (Khalizov et al., 2009b;Pagels et al., 2009), which ultimately enhance BC aging. It is provided here that in macroscopic view BC aging and subsequent removal result in variation of ΔBC/ΔCO values and correlation coefficients between BC and CO, which deepens the comprehensive understanding on BC aging.**

We also thank the editor for the careful and kind help with editing the English and have already examed the entire text to improve the readability.

**A list of all relevant changes made in the manuscript**

1. Line 28 - 29 in Page 1 and Line 1 - 2 in Page 2

2. Line 5 - 9 in Page 7

[revised manuscript text omitted]